# Effects of Different Cutting Widths on Physical and Mechanical Properties of Moso Bamboo under Strip Cutting

Liyang Liu [1,2], Xiao Zhou [1,2], Zhen Li [1,2], Xuan Zhang [1,2] and Fengying Guan [1,2,*]

1  International Center for Bamboo and Rattan, Key Laboratory of National Forestry and Grassland Administration, Beijing 100102, China
2  National Location Observation and Research Station of the Bamboo Forest Ecosystem in Yixing, National Forestry and Grassland Administration, Yixing 214200, China
*  Correspondence: guanfy@icbr.ac.cn; Tel.: +86-10-84789808

**Abstract:** We aimed to explore the effect of strip cutting width on the physical and mechanical properties of Moso bamboo (*Phyllostachys edulis*). Four-year-old hairy bamboo with different cutting bandwidths (3 m, 5 m, 8 m, 12 m and 15 m) was used as the experimental material, and the traditional management forest (CK) was used as the control. Eight physical and mechanical properties—radial line shrinkage, chordal line shrinkage, volume shrinkage, total dry density, basic density, compressive strength, shear strength and flexural strength—were studied. ANOVA, bivariate correlation analysis, and principal component analysis were performed, which showed the effect of strip cutting on the physical and mechanical properties of bamboo. The results showed that the density, the total dry density, basic density, flexural strength and compressive strength of the strip cut bamboo were lower than those of the control stand. The radial line shrinkage and volume shrinkage were higher than those of the control stand. The chordal line shrinkage was higher than that of the control stand when the cutting bandwidth was 3 m and 5 m and lower than that of the control stand when the cutting bandwidth was 8 m, 12 m and 15 m. The shear strength was lower than that of the control stand when the harvesting bandwidth was 3 m, 5 m and 8 m and higher than that of the control stand when the harvesting bandwidth was 12 m and 15 m. There are different degrees of correlation between the eight physical and mechanical performance indicators. The comprehensive score of the physical and mechanical properties of bamboo in the control stand was 1.30, and the comprehensive score of bamboo in strip harvesting was between 0.37 and 1.25, with an average score of 0.95. The results show that strip cutting can save the time and costs associated with harvesting Moso bamboo forests, but different cutting widths can reduce the physical and mechanical properties of Moso bamboo in different degrees.

**Keywords:** Moso bamboo; strip logging; physical properties; mechanical properties





## 1. Introduction

Moso bamboo (*Phyllostachys edulis*) is the oldest cultivated bamboo species in China, with a wide range of uses and high ecological and economic value [1]. Its area accounts for 70% of the total area of bamboo forests in China and 80% of the total area of bamboo forests in the world [2]. Because Moso bamboo forest is a typical forest with different ages and shoots and grows bamboo every year or every other year, the traditional selective cutting operation has been used as the cutting method for Moso bamboo forests. However, selective cutting can only be carried out manually, which greatly reduces the benefits of bamboo resource cultivation and limits the enthusiasm of bamboo farmers for bamboo forest management [3]. Based on the annual growth of bamboo demand, improving the efficiency of collecting materials and reducing the cost of harvesting have become a difficult problem in the management of bamboo. Some scholars proposed a highly mechanized harvesting method—strip harvesting based on the complex underground

whip root system and clonal integration characteristics of Moso bamboo [4]. We aimed to explore the feasibility of a mechanized harvesting mode for bamboo forests to help the transformation and upgrading of the bamboo industry.

The quality of bamboo structural materials such as bamboo wood-based boards and bamboo furniture directly depends on the physical and mechanical properties of bamboo [5]. Chung et al. studied the mechanical properties of Moso bamboo used as a building material, and the results showed that Moso bamboo has excellent compressive strength and flexural strength, which makes it a good structural material [6]. The physical and mechanical properties of bamboo mainly include moisture content, density, dry shrinkage, compressive strength, flexural strength, flexural elastic modulus, etc., which are very important for the processing and utilization of bamboo [7]. When used as a structural material, excellent dimensional stability and mechanical properties are usually required, which requires small dry shrinkage and high flexural, compressive and shear strength in bamboo. Panshin summarized a large number of research results on wood material variability and pointed out that wood material variation comes from three factors: age, environment (such as altitude, site conditions, etc.) and genetic structure. The same is true for bamboo [8]. In this study, the age and origin of the bamboo were the same, so the site condition may have been the main influencing factor on the difference in physical and mechanical properties. In the study of strip cutting, many scholars found that strip cutting can improve soil quality and help to restore the soil fertility of Moso bamboo forest [9–11]. The width of small and medium cutting is significantly promoted, and improvements in soil nutrient content will accelerate the growth of bamboo. Some studies have found that better site conditions will lead to a decline in material quality [12,13]. According to previous studies, when trees grow fast, the fiber length of wood in thinning forests increases significantly [14], and fiber morphology is an important factor affecting the quality of wood products [15]. Density is also an important parameter of bamboo properties, and the difference in density will affect the mechanical properties.

At present, the research on the physical and mechanical properties of bamboo mainly focuses on the differences between bamboo of different ages or different parts. Yu et al. believed that the mechanical properties of bamboo with 3~5 degrees were stable at a high level, and then the bamboo would be aged [16]. Li et al. believed that the physical and mechanical properties of three-year-old bamboo were significantly different from those of two-year-old bamboo, but there was little difference in bamboo born after three years [17]. Zhang et al. believed that the mechanical properties of 4-year-old bamboo were generally better than those of 6-year-old bamboo [18]. Li found that the air-dry density, flexural strength, flexural elastic modulus and flexural compressive strength of bamboo gradually increased from the base to the tip [17]. Zhou found that the density of the vascular bundles of the same bamboo species decreases with the increase in the height and diameter of the bamboo breast [19]. As the density of the vascular bundles decreases, the public bulk density and mechanical strength of the bamboo also decrease.

According to different terrain, strip cutting is carried out on the stands, and retention plots of the same width are set on both sides of the stands. Nutrients are transported through the underground whip root system to support the growth and development of Hsinnex in the stands. So, strip cutting is different from traditional selective cutting, which can obtain Moso bamboo with uniform ages, and strip cutting can harvest Moso bamboo with different ages. Scholars have carried out a number of studies on biomass [20,21], soil quality [10,22], soil nutrients [23,24], Hsinchu quantity [20,25] and understory vegetation diversity [26]. The changes in bamboo properties after strip cutting have not been reported, so there is no theoretical guidance for the utilization of these bamboo with different cutting widths and ages. Paying attention to the differences of bamboo properties in different cutting widths and bamboo ages can provide a basis for the subsequent processing and utilization of bamboo and is of great significance for the promotion of strip cutting.

In this study, the density, dry shrinkage, flexural strength, compressive strength and shear strength of Moso bamboo wood with different harvesting bandwidths under strip

cutting were studied; Moso bamboo from a traditional management forest (CK) was used as a control, and the effect of strip cutting on physical and mechanical properties was clarified, which provided a reference for the utilization of Moso bamboo forest after strip cutting.

## 2. Materials and Methods

### 2.1. Test Site Profile

The study was located in the east of Taihu Lake in Yixing City, Jiangsu Province, China. The experimental plot was set in Yixing State-owned forest farm ($31°15'1''\sim31°15'40''$ N, $119°43'52''\sim119°44'41''$ E), adjacent to Longchi Mountain, which is the border of the northern scattered bamboo area and the Jiangnan bamboo area, and the northeast edge of the distribution area of Moso bamboo (Figure 1).

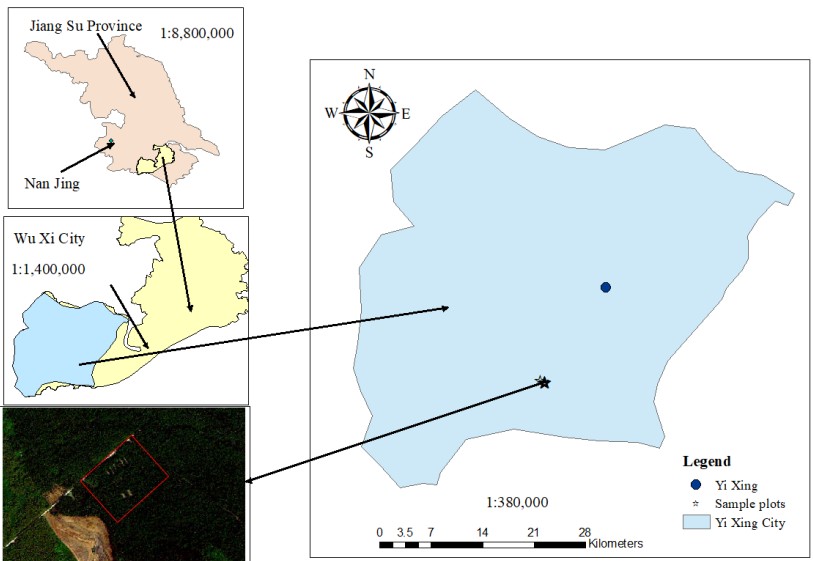

**Figure 1.** Location of the study area.

### 2.2. Experimental Design

In the winter of 2017, the centralized contiguous pure Moso bamboo forests with consistent stand conditions were selected for strip-like clear-cutting. Five strip harvesting plots were set, and the widths of the cutting strips were 3 m, 5 m, 8 m, 12 m and 15 m, and the lengths were all 20 m. Retention strips of the same width were set between the harvesting strips, and trenches of 40 cm × 50 cm were dug around each of the five strip-shaped harvesting plots for isolation, as shown in Figure 2. At the same time, three 20 m × 20 m control plots without logging were set up around the cutting sample plots to implement traditional management. To ensure similar areas among the harvested plots, plots with different harvesting widths had different numbers (*n*) of harvesting zones; the serial numbers were 1, 3, 5... The basic information of the sample plot is shown in Table 1.

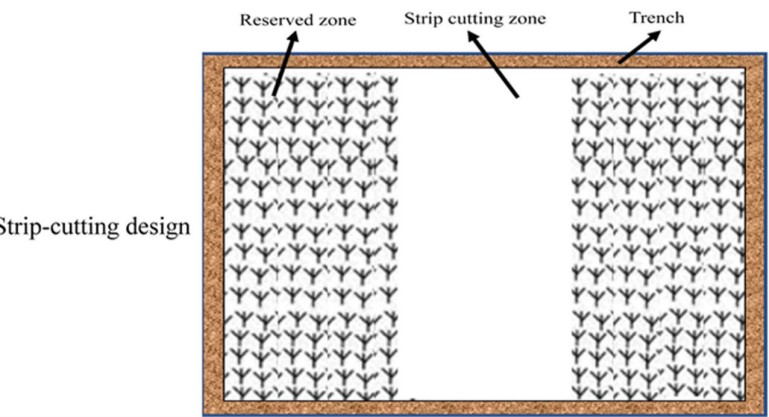

**Figure 2.** Schematic diagram of strip cutting. The cutting width is 3 m, 5 m, 8 m, 12 m or 15 m, and the cutting length is 20 m.

**Table 1.** Basic characteristics of the three sampling plots.

| Code | Areas (m$^2$) | Number (n) | Longitude | Latitude | Altitude (m) | Slope (°) |
|---|---|---|---|---|---|---|
| 3 m | 20 × 3 | 10 | 119°44′6″ E | 31°15′1″ N | 166.43 | 2 |
| 5 m | 20 × 5 | 6 | 119°44′6″ E | 31°15′1″ N | 165.55 | 2 |
| 8 m | 20 × 8 | 4 | 119°44′6″ E | 31°15′1″ N | 164.29 | 2 |
| 12 m | 20 × 12 | 3 | 119°44′43″ E | 31°15′37″ N | 159.59 | 8 |
| 15 m | 20 × 15 | 3 | 119°44′47″ E | 31°15′38″ N | 158.39 | 8 |
| CK | 20 × 20 | 3 | 119°44′6″ E | 31°15′1″ N | 167.10 | 2 |

*2.3. Research Methods*

2.3.1. Material Collection

Three 4-year old standard bamboo trees were selected from the zone-shaped harvesting plots (the harvesting zones numbered 1, 3 and 5 were selected for different cutting widths) and the traditional management forest (CK). After leveling, a section about 2.0 m long was cut up from the whole bamboo node about 1.5 m above the ground and cut off at the whole bamboo node to be used as test materials.

2.3.2. Experimental Method

The following bamboo tube was taken from the 2 m bamboo section, and one bamboo strip of more than 15 mm and 30 mm was split in the southeast and northwest directions, respectively. Then, 15 mm bamboo strips were used to make dry shrinkage, density and flexural strength specimens. The 30 mm bamboo strip was used to make the compression and shear samples along the grain. The test section used for dry shrinkage samples needed to be soaked in clean water at room temperature until the size was basically stable before the sample was made, and the other test strips needed to be dried in the room after the gas drying. After making the sample with the air-dried test strip, it was placed in a constant-temperature and constant-humidity box with the temperature of 20 ± 2 °C and the relative humidity of 65% ± 5%, and the moisture content of the sample was adjusted to the equilibrium state of about 12%. The dry shrinkage and density were measured along the grain compressive strength; flexural strength was measured along the grain shear strength according to GB/T 15780-1995 [27]. The instrument used was the electronic universal testing machine (Instron manufacturer, 5582, Boston, MA, USA).

2.3.3. Analysis Methods

SPSS 26.0 was used to analyze the physical and mechanical properties via multivariate analysis of variance (MANOVA), and a principal component analysis (PCA) (KMO and

Bartlett's sphericity test were used for correlation analysis to select suitable samples for PCA) was performed to comprehensively evaluate the physical and mechanical properties of the strip harvesting of Moso bamboo. Bivariate correlation analysis was performed using R (version 4.2.2) to analyze the correlation between eight physical and mechanical performance indicators.

The implementation steps of comprehensive evaluation are as follows:

(1)　Data standardization (range method):

Positive index (density, flexural strength, compressive strength, shear strength):

$$Y_{ij} = \frac{X_{ij} - \min(X_{ij})}{\max(X_{ij}) - \min(X_{ij})}$$

Negative index (dry shrinkage):

$$Y_{ij} = \frac{\min(X_{ij}) - X_{ij}}{\max(X_{ij}) - \min(X_{ij})}$$

(2)　Analysis: Based on standardized data, principal component analysis was performed in spss to obtain comprehensive scores.

(3)　Score: According to the 3σ principle in statistics, we used the formula

$$Y_i^t = H + Y_i$$

Coordinate translation was performed to eliminate the negative influence.

## 3. Results

### 3.1. Differences in Physical Properties

#### 3.1.1. Density

Density is one of the important material properties, which is related to mechanical strength. Figure 3 shows the variation in the total dry density and basic density of Moso bamboo with harvesting bandwidth under strip harvesting. The change trend of total dry density was consistent with that of basic density, which decreased first and then increased with the increase in harvesting bandwidth. The total dry density of strip cut bamboo was between 0.56 g/cm$^3$ and 0.68 g/cm$^3$, which was lower than that of the control stand, and it reached the lowest at 8 m, followed by 5 m, and there was no significant difference between 3 m, 5 m, 12 m and 15 m. The basic density ranged from 0.51 to 0.61 g/cm$^3$, which was lower than that of the control stand, and it reached the lowest when the harvesting bandwidth was 8 m, and the difference in the rest of the harvesting bandwidth was not significant.

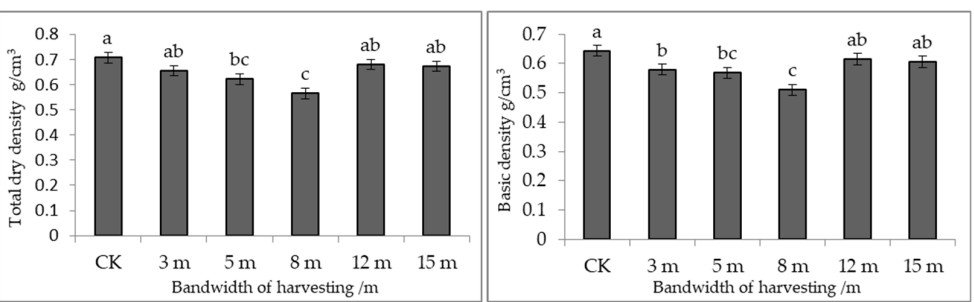

**Figure 3.** Density of Moso bamboos under strip harvesting. Note: Different letters indicate a signifcant difference in density in different treatments ($p < 0.05$).

#### 3.1.2. Dry Shrinkage

Figure 4 shows the variation in the dry shrinkage of bamboo under strip harvesting as a function of harvesting bandwidth. The radial line shrinkage and chordal line shrinkage

had the same change trend. With the increase in harvesting bandwidth, the linear shrinkage first decreased, then slightly increased and then decreased, and the volume shrinkage first increased, then decreased, and then stabilized. The radial line shrinkage was 4.6%–7.3%, which was higher than that of the control stand, and reached the highest value when the cutting bandwidth was 3 m, which was significantly higher than that of the control stand, but there was no significant difference between 5 m, 8 m, 12 m and 15 m. The chordal line shrinkage ranged from 0.49% to 1.9%; 3 m and 5 m were higher than the control stand, 8 m, 12 m and 15 m were lower than the control stand, and the cutting bandwidth reached the highest value at 3 m, which was significantly higher than that of the control stand. There was no significant difference in the other cutting bandwidth. The volume shrinkage ranged from 9.5% to 13.4%, which was higher than that of the control stand.

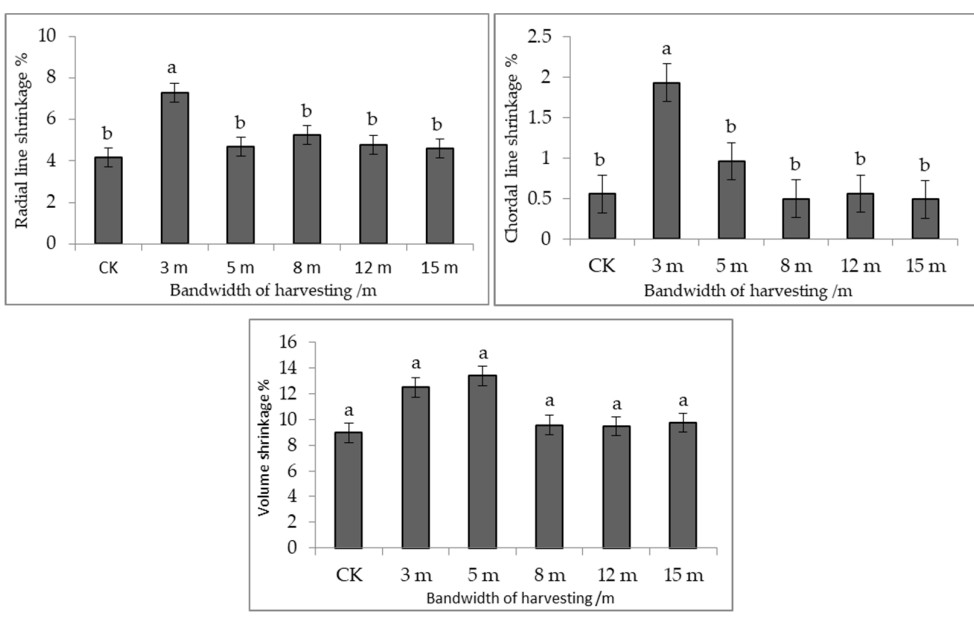

**Figure 4.** Dry shrinkage of Moso bamboo under strip cutting. Note: Different letters indicate a signifcant difference in shrinkage in different treatments ($p < 0.05$).

*3.2. Mechanical Properties*

3.2.1. Flexural Strength

Figure 5 shows the flexural strength of bamboo under strip harvesting as a function of harvesting bandwidth. The flexural strength decreases first and then increases with the increase of harvesting bandwidth. Compared with the control stand, the flexural strength was reduced by strip cutting, and the flexural strength was significantly reduced by 3 m, 5 m and 8 m cutting bandwidths, which were reduced by 13.0%, 19.6% and 26.2%, respectively. There was no significant difference in the harvesting bandwidth of 12 m and 15 m, which was reduced by 5.5% and 0.3%, respectively.

3.2.2. Compressive Strength

Figure 6 shows the variation in the compressive strength of Moso bamboo with harvesting bandwidth. With the increase in harvesting bandwidth, the compressive strength showed a trend of first decreasing, then increasing and then decreasing slightly. Compared with the control stand, the compressive strength was reduced by strip cutting, and the bending strength was significantly decreased by 8.3%, 19.5% and 25.0% in 3 m, 5 m and 8 m cutting bandwidths, respectively. There was no significant difference in the harvesting bandwidth of 12 m and 15 m, which was reduced by 4.2% and 5.1%, respectively.

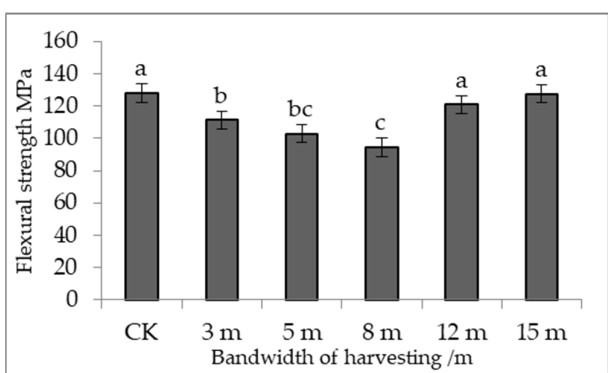

**Figure 5.** Flexural strength of bamboo under strip cutting. Note: Different letters indicate a signifcant difference in flexural strength in different treatments ($p < 0.05$).

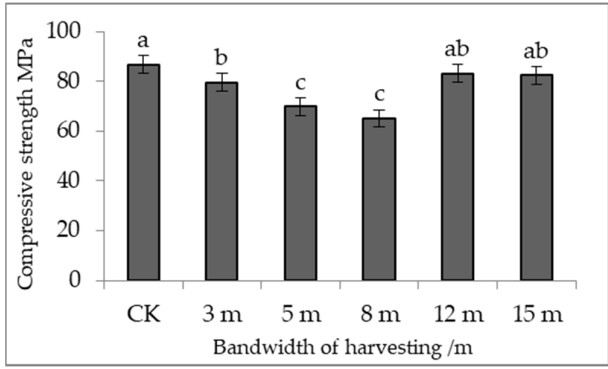

**Figure 6.** Compressive strength of Moso bamboo under strip cutting. Note: Different letters indicate a significant difference in compressive strength in different treatments ($p < 0.05$).

### 3.2.3. Shear Strength

Figure 7 shows the shear strength of Moso bamboo under strip harvesting as a function of harvesting bandwidth. The shear strength decreased first and then increased with the increase in harvesting bandwidth. Compared with the control stand, the shear strength of 5 m and 8 m cutting bandwidth was significantly reduced by 14.4% and 12.2%, respectively. The harvesting bandwidth of 3 m, 12 m, and 15 m was not significantly different; 3 m was reduced by 6.4%, 12 m was increased by 1.9% and 15 m was increased by 6.4%.

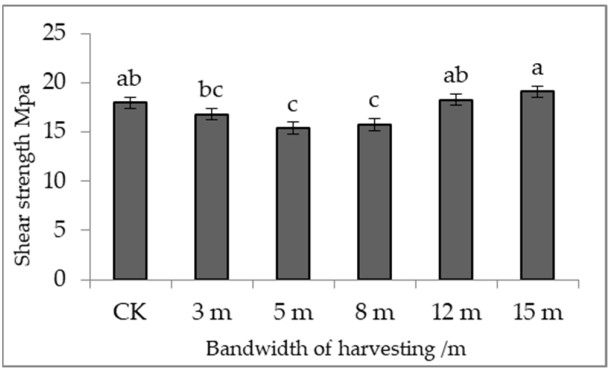

**Figure 7.** Shear strength of Moso bamboo under strip cutting. Note: Different letters indicate a signifcant difference in shear strength in different treatments ($p < 0.05$).

### 3.3. Correlation Analysis

The analysis of the correlation between physical and mechanical properties is helpful to consider the changes in related properties when selecting building materials. The results

of the correlation analysis of eight indicators of physical and mechanical properties of Moso bamboo showed that there were different degrees of correlation among the indicators (Figure 8). There was a strong correlation between radial line shrinkage and volume shrinkage, between total dry density and basic density, flexural strength and compressive strength, between basic density and flexural strength and compressive strength and between flexural strength and compressive strength, and the correlation coefficients were 0.62, 0.74, 0.63, 0.65, 0.83, 0.78 and 0.65, respectively. There was a moderate correlation between volume shrinkage and total dry density, between basic density and shear strength, between flexural strength and shear strength and between compressive strength and shear strength, and the correlation coefficients were 0.49, 0.44, 0.44 and 0.47, respectively. There was a weak correlation between total dry density and shear strength with a correlation coefficient of 0.29. There was a weak negative correlation between radial line shrinkage and basic density, and the correlation coefficient was −0.24. The remaining indicators were very weakly correlated or had no correlation with each other.

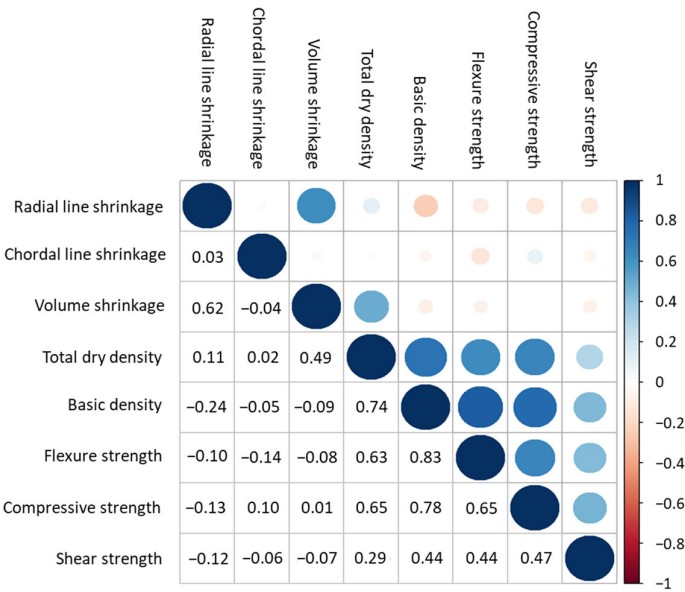

**Figure 8.** Correlation analysis results.

*3.4. Comprehensive Evaluation*

The KMO test results of eight physical and mechanical properties of bamboo under strip harvesting were 0.641, and the significance coefficient of Bartlett spherical test was 0.00 (<0.05), indicating that the data were suitable for principal component analysis. According to the results of principal component analysis (Table 2), four components were extracted, and the cumulative contribution rate was 85.69%. The eigenvalues of the first four components were 3.044, 1.979, 0.938 and 0.671, and the variance contribution rates were 38.046%, 24.744%, 11.726% and 8.384%, respectively.

Therefore, the four principal components were used as comprehensive variables to evaluate the physical and mechanical properties of Moso bamboos after strip harvesting. According to Table 3, the first principal component reflects the total dry density, basic density, flexural strength and compressive strength. The second principal component reflects radial line shrinkage and volume shrinkage. The third principal component reflects the shear strength. The fourth principal component reflects chordal line shrinkage.

**Table 2.** Eigenvalues and contribution rates of principal components of physical and mechanical properties of Moso bamboos.

| Ingredient | Initial Eigenvalue | | | Extract Sum of Squares and Load | | |
|---|---|---|---|---|---|---|
| | Eigenvalue | Contribution Rate % | Accumulating Contribution Rate % | Eigenvalue | Contribution Rate % | Accumulating Contribution Rate % |
| 1 | 3.311 | 41.383 | 41.383 | 3.311 | 41.383 | 41.383 |
| 2 | 1.980 | 24.749 | 66.132 | 1.980 | 24.749 | 66.132 |
| 3 | 0.908 | 11.346 | 77.478 | 0.908 | 11.346 | 77.478 |
| 4 | 0.657 | 8.207 | 85.685 | 0.657 | 8.207 | 85.685 |
| 5 | 0.511 | 6.387 | 92.072 | | | |
| 6 | 0.331 | 4.140 | 96.212 | | | |
| 7 | 0.246 | 3.078 | 99.29 | | | |
| 8 | 0.057 | 0.710 | 100 | | | |

**Table 3.** Physical and mechanical property index of principal component eigenvectors.

| Target | Principal Component Eigenvectors | | | | Communality | Weight/$a_i$ |
|---|---|---|---|---|---|---|
| | PC-1 | PC-2 | PC-3 | PC-4 | | |
| Radial line shrinkage | 0.139 | 0.530 | −0.128 | −0.214 | 0.740 | 0.108 |
| Chordal line shrinkage | −0.005 | −0.117 | −0.092 | 0.979 | 0.971 | 0.142 |
| Volume shrinkage | −0.006 | 0.475 | 0.060 | 0.039 | 0.857 | 0.125 |
| Total dry density | 0.282 | −0.230 | −0.086 | −0.079 | 0.912 | 0.133 |
| Basic density | 0.445 | 0.136 | −0.361 | 0.004 | 0.780 | 0.114 |
| Flexural strength | 0.332 | 0.060 | −0.097 | 0.114 | 0.806 | 0.118 |
| Compressive strength | 0.217 | 0.022 | 0.211 | −0.088 | 0.826 | 0.121 |
| Shear strength | −0.241 | −0.033 | 1.032 | −0.063 | 0.962 | 0.140 |

The comprehensive evaluation results of physical and mechanical properties showed that the comprehensive score of hairy bamboo in the control stand was 1.30, and the average score of hairy bamboo under strip harvesting was 0.37 to 1.25, and the average score was 0.95. The score of hairy bamboo in 3 m harvesting bandwidth was the highest, followed by 15 m, 12 m, 5 m and 8 m harvesting bandwidth, which was the lowest.

## 4. Discussion

Panshin [8] summarized a large number of research results on wood material variability and pointed out that wood material variation came from three factors: age, environment (such as altitude, site conditions, etc.), and genetic structure. By changing the composition of forest trees and system energy distribution, logging will affect the microenvironment and physical and chemical properties of forest soil [28]. In this study, the age and origin of bamboo were the same, so site conditions and bamboo standing degree may have been the main influencing factors for the differences in physical and mechanical properties. Regarding the effect of harvesting on wood properties, several scholars have focused on the effect of thinning on the physical and mechanical properties of wood. Krajnc, L. [29] and Chen Ruiying [30] believed that thinning would have a negative impact on wood material properties; Luo Zhenfu [31] believed that 45% thinning could not only significantly accelerate the growth of pine, but also improve the mechanical properties; Chen Guangsheng [32] believed that moderate thinning could not only improve the mechanical properties of wood, but also accelerate the growth of trees.

In this study, we analyzed the differences in the physical and mechanical properties of Moso bamboo under strip cutting with different cutting widths and control stands. With the increase in harvesting bandwidth, the density, flexural strength and shear strength of Moso bamboo first decreased and then increased, and the compressive strength first decreased and then increased and then decreased slightly, and the density, flexural strength

and compressive strength of Moso bamboo in each cutting width were lower than those of the control stand. The shear strength was lower than that of the control stand when the harvesting bandwidth was 3 m, 5 m and 8 m, and it was higher than that of the control stand when the harvesting bandwidth was 12 m and 15 m. Wang Shumei et al. [9,10] found that the small and medium cutting width of strip harvesting was conducive to the improvement in soil fertility, while the large cutting width hindered nutrient cycling. Better site conditions not only affect the growth rate of Moso bamboo, but also cause a decline in the mechanical properties of bamboo [12], especially on the flexural modulus, tensile elastic modulus and maximum compression force. When the growth is rapid, the vertical division speed of cambium primitive cells is faster, the space of transport tissue is large and the cell cavity is large and thin, resulting in a slightly loose wood texture [14]. With the increase in strip cutting width, the Hsinchu quantity first increased and then decreased [9,25]. In higher-density stands, trees compete strongly for soil nutrients and water, and the conditions of light and air ventilation are poor, so the vitality of cambium is greatly weakened. As the meristem ability of cambium primitive cells is negatively correlated with the length of newly formed daughter cells [8], the length of fiber cells in the strip harvesting of bamboo wood decreases first and then increases with the increase in harvesting width. Fiber length is thought to be positively correlated with flexural strength, but not to the same extent as density [14]. The radial dry shrinkage and volume dry shrinkage of strip cut bamboo were higher than those of the control stand. The chordwise dry shrinkage was higher than the control stand at 3 m and 5 m harvesting bandwidths and lower than the control stand at 8 m, 12 m and 15 m harvesting bandwidths. The shrinkage of radial line and chordal line decreased first, then increased slightly, and then decreased. The volume shrinkage first increased, then decreased, and then stabilized. The dry shrinkage and density showed an opposite change trend. When the bamboo material was tight, the dry shrinkage of bamboo decreased, which is consistent with the research results of Gao Shanshan [33] and Ren Shiqi [34].

## 5. Conclusions

In this study, it was shown that the strip harvesting of Moso bamboo may lead to a decrease in physical and mechanical properties due to the increase in soil nutrients. The total dry density of strip cut bamboo was between 0.56 g/cm$^3$ and 0.68 g/cm$^3$, and the order was ck > 12 m > 15 m > 3 m > 5 m > 8 m. The basic density was between 0.51 and 0.61 g/cm$^3$, and the order was ck > 12 m > 15 m > 3 m > 5 m > 8 m. The dry shrinkage of the radial line was between 4.6% and 7.3%, and the order was 3 m > 8 m > 5 m > 12 m > 15 m > ck. The chordal linear shrinkage was between 0.49% and 1.9%, and the order was 3 m > 5 m > 12 m > ck > 8 m > 15 m. The volume dry shrinkage was between 9.5% and 13.4%, and the order was 5 m > 3 m > 15 m > 12 m > 8 m > ck. The bending strength was between 94.4 Mpa and 127.6 Mpa, and the order was ck > 15 m > 12 m > 3 m > 5 m > 8 m. The compressive strength was between 65.1 Mpa and 73.1 Mpa, and the order was ck > 12 m > 15 m > 3 m > 5 m > 8 m. The shear strength was between 15.4 Mpa and 19.1 Mpa, and the order was 15 m > 12 m > ck > 3 m > 8 m > 5 m. The total dry density, basic density, flexural strength and compressive strength were lower than those of the control stand. The radial dry shrinkage and volume dry shrinkage were higher than those of the control stand. The chordwise dry shrinkage was higher than that of the control stand when the cutting bandwidth was 3 m and 5 m, and lower than that of the control stand when the cutting bandwidth was 8 m, 12 m and 15 m. The shear strength was lower than that of the control stand when the harvesting bandwidth was 3 m, 5 m and 8 m, and higher than that of the control stand when the harvesting bandwidth was 12 m and 15 m. The average score of physical and mechanical properties was 0.95, which was lower than that of the control stand, and the order of score was ck > 3 m > 15 m > 12 m > 5 m > 8 m.

*Outlook*

At present, there is no report on timber properties of strip harvesting. The anatomical properties are closely related to the physical and mechanical properties. With regard to the reasons for the decline in the physical and mechanical properties of Moso bamboo, it would be better to conduct further research on the anatomical characteristics of Moso bamboo, so as to verify the discussion section in this paper. At the same time, focusing on the changes in bamboo properties at different ages will also help the utilization of bamboo in strip harvesting.

**Author Contributions:** Conceptualization, F.G.; methodology, L.L.; software, X.Z. (Xiao Zhou); validation, X.Z. (Xuan Zhang); formal analysis, Z.L.; investigation, L.L.; resources, X.Z. (Xiao Zhou); data curation, Z.L.; writing—original draft preparation, L.L.; writing—review and editing, X.Z. (Xuan Zhang); visualization, X.Z. (Xiao Zhou); supervision, F.G.; project administration, F.G.; funding acquisition, F.G. All authors have read and agreed to the published version of the manuscript.

**Funding:** Basic Scientific Research Funding of International Center for Bamboo and Rattan (1632023013). The 14th Five-Year National Key Research and Development Plan Project "Research on Multi-objective Precise and Efficient Cultivation Technology of Bamboo and Rattan Resources" (SQ2023YFD2200013).

**Data Availability Statement:** The data that support the findings of this study are available from the corresponding author upon reasonable request.

**Acknowledgments:** Thanks to several authors for their help and the Yixing forest farm staff for their support.

**Conflicts of Interest:** The authors declare no conflict of interest.

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
