# Peer review of "Effects of Different Cutting Widths on Physical and Mechanical Properties of Moso Bamboo under Strip Cutting"

_forests, doi:10.3390/f14102068_

Round 1

Reviewer 1 Report

Effects of different cutting widths on physical and mechanical properties of Bamboos pilosa under strip cutting

Habitat conditions and growth conditions significantly affect the properties of wood raw material obtained from different areas. The conditions for the growth of subsequent newly planted trees may depend on the method of felling the previous trees. In this context, the subject of the article can be considered interesting. Nevertheless, the authors did not fully explain how their research was conducted. They studied the influence of the cutting width (harvesting) of bamboos on their physical and mechanical properties. 6 cutting widths were used on plots specially selected for this purpose in 2017. However, the authors did not sufficiently explain the differences in the growth conditions of individual bamboo groups:

·         Were cuttings of the assumed width carried out on the plots that constituted the research area in 2017, and then new bamboo seedlings were planted in this place, which in turn were cut down after four years and samples were taken for testing?

·         Or were cuts of the assumed width made in a forest that grew under the same conditions? In this case, however, how could the width of the bamboo cut affect their properties?

Therefore, the authors should explain in more detail the purpose and method of preparing the raw materials for testing.

In addition, the authors should strictly separate the discussion of the results from the conclusions.

Author Response

1. Summary

Thank you very much for taking the time to review this manuscript. Please find the detailed responses below and the corrections highlighted in the re-submitted files.

2. Point-by-point response to Comments and Suggestions for Authors

Comments 1: Were cuttings of the assumed width carried out on the plots that constituted the research area in 2017, and then new bamboo seedlings were planted in this place, which in turn were cut down after four years and samples were taken for testing?

Response 1:

In the study of bamboo, bamboo seedlings are not generally planted. After harvesting, the whip root grows into bamboo shoots and develops into Hsinchu. Therefore, we did not consider genetic factors (which may be what you mean by individual differences) in the case of consistent provenance.

Comments 2: Or were cuts of the assumed width made in a forest that grew under the same conditions? In this case, however, how could the width of the bamboo cut affect their properties?

Response 2: The stand conditions selected in this study are consistent (L113) . At the beginning of strip harvesting, our results show that the soil is greatly disturbed after harvesting, and the change of growth conditions of Moso bamboo is easy to cause changes in wood properties (I added this part in the introduction :L60-L71).

Comments 3: Therefore, the authors should explain in more detail the purpose and method of preparing the raw materials for testing.

Response 3: Thank you for pointing this out. We agree with this comment. Therefore, I add in the introduction and research methods (L100、L121-122、L130-132、L143-146) .

Comments 4: In addition, the authors should strictly separate the discussion of the results from the conclusions.

Response 4: Thank you for pointing this out. We agree with this comment. I separate the discussion from the conclusion and quote some articles on the effect of harvesting on wood quality.

Reviewer 2 Report

The paper presents a statistical investigation to discover the effect of a special harvesting technique, the strip cutting, on eight physical-mechanical properties of bamboo.

Major remarks

1. the strip cutting is not described at all (and the provided references are not sufficiently useful), so that the rationale of the research is obscure to the reader! Surprisingly, the Conclusion and Discussion section (L245-250; L281-285) gives some (vague) information on the question.
This section should be moved at the Introduction; furthermore, it should be sufficiently expanded, to offer a solid motivation of the paper.

2. the specie of the bamboo investigated is named in several different ways in the paper (Line 2: pilosa, L39: Moso/phyllostachys, L166: chinensis, L179: phyllostellate, L197: phyllostachys bambotum, L259: phyllostaphus pilosa;

3. Figs. 3-7: The different bandwidths appear to do not have significant influence on the investigated quantities. In other words: the KMO= 0.64 synthetizes the impression that the correlations investigated seem to be rather weak so that the factor analysis appears to be scarcely significant (except for almost trivial connections, e.g. between compression and flexural strength, density and strength, and similar).

4. It is not clear the number of samples involved in the statistical investigation: per area (cutting plot), per each physical or mechanical test, ... In particular: Number (n) in Table 1 is not defined. This information is crucial for the reader.

5. It is not discussed why the values of the control CK (not defined in the manuscript!) are (almost) always greater than (or seldom clearly lesser) of the samples investigated. The value of the paper greatly depends on this explanation.

Minor remarks

6. L12: Please consider omitting the “including” term.

7. L31: Please consider moving the correlation coefficient close to the relevant quantities.

8. L35: Please define CK

9. L67: “K.f. hung” is perhaps “KF Chung”

10.  Please include in References some of the numerous studies of the mechanical properties, available in the international literature.

11. L99: Are “Cutting width” and “Bandwidth” equivalent terms?

12. The numbers 3 in the areas 15m e CK seems to be excessive, with respect to the previous areas (10x3, 5x6, 8x4, 12x3). Please explain.

13. L111. “The 30mm bamboo strip is used to make the compression and shear... ” Is 30 mm the dimension of the samples in compression ?!

Author Response

For research article

Response to Reviewer Comments

1. Summary

Thank you very much for taking the time to review this manuscript. Please find the detailed responses below and the corrections highlighted in the re-submitted files.

2. Point-by-point response to Comments and Suggestions for Authors

Comments 1: 1. the strip cutting is not described at all (and the provided references are not sufficiently useful), so that the rationale of the research is obscure to the reader! Surprisingly, the Conclusion and Discussion section (L245-250; L281-285) gives some (vague) information on the question.

This section should be moved at the Introduction; furthermore, it should be sufficiently expanded, to offer a solid motivation of the paper.

Response 1: Thank you for pointing this out. We agree with this comment. Therefore, I make some additions in the introduction section. The method of strip cutting and the factors that may cause the change of timber property are described(L60-71、L86-89). (l245 - 250in the original manuscript) moved to the introduction.

Comments 2: the specie of the bamboo investigated is named in several different ways in the paper (Line 2: pilosa, L39: Moso/phyllostachys, L166: chinensis, L179: phyllostellate, L197: phyllostachys bambotum, L259: phyllostaphus pilosa;

Response 2: Thank you for pointing this out. We agree with this comment. I made a change(L3、154、170、180、202、215、226、228、235、264、274、348、350)

Comments 3: The different bandwidths appear to do not have significant influence on the investigated quantities. In other words: the KMO= 0.64 synthetizes the impression that the correlations investigated seem to be rather weak so that the factor analysis appears to be scarcely significant (except for almost trivial connections, e.g. between compression and flexural strength, density and strength, and similar).

Response 3: As for kmo=0.64, I read relevant literature before writing, and finally found the basis. KMO is one of the validity test indicators for principal component analysis. It is written in the previous literature that KMO is above 0.9, which is very suitable for factor analysis. Between 0.8 and 0.9, which is very suitable; Between 0.7-0.8, suitable; Between 0.6-0.7, OK; A value between 0.5 and 0.6 is very poor. Values below 0.5 should be abandoned (Kaiser 1974).

Comments 4: It is not clear the number of samples involved in the statistical investigation: per area (cutting plot), per each physical or mechanical test, ... In particular: Number (n) in Table 1 is not defined. This information is crucial for the reader.

Response 4: Thank you for pointing this out. We agree with this comment. I modified it accordingly (L121-122、130-132).

Comments 5: It is not discussed why the values of the control CK (not defined in the manuscript!) are (almost) always greater than (or seldom clearly lesser) of the samples investigated. The value of the paper greatly depends on this explanation.

Response 5: CK is a sample plot of Moso bamboo forest with traditional management, which serves as a control. The change of physical and mechanical properties of bamboo after strip cutting was illustrated by comparing it with strip cutting. I added an introduction to CK (L12-13、100、130-132).

Comments 6: Please consider omitting the “including” term.

Response 6: I corrected that (L12).

Comments 7: L31: Please consider moving the correlation coefficient close to the relevant quantities.

Response 7: I have removed this section to simplify the summary.

Comments 8: L35: Please define CK

Response 8: I added that (L12-13、100).

Comments 9: L67: “K.f. hung” is perhaps “KF Chung”

Response 9: I corrected that (L50).

Comments 10: Please include in References some of the numerous studies of the mechanical properties, available in the international literature.

Response 10: According to the content modification, I modified the references.

Comments 11: L99: Are “Cutting width” and “Bandwidth” equivalent terms?

Response 11: Yes.

Comments 12: The numbers 3 in the areas 15m e CK seems to be excessive, with respect to the previous areas (10x3, 5x6, 8x4, 12x3). Please explain.

Response 12: In order to ensure that the strip logging plots (fig 2: logging plots are composed of strip cutting zones, reserved zones and trench) are similar in area(L120-122). The material used for the study was taken from the cutting strip (L130).

Comments 13: L111. “The 30mm bamboo strip is used to make the compression and shear... ” Is 30 mm the dimension of the samples in compression ?!

Response 13: Yes. 30mm bamboo strips are used to make compressive and shear resistant specimens.

Reviewer 3 Report

Dear Authors

I had the opportunity to review (11 pages) research proposed for Forests under the name "Effects of different cutting widths on physical and mechanical properties of Bamboos pilosa under strip cutting". The article deals with an interesting topic which was focused on the effect of strip cutting width on the physical and mechanical properties of bamboo. Eight physical and mechanical properties for example radial shrinkage, volume shrinkage, total dry density, basic density, compressive strength, and flexural strength were chosen. Four-year-old bamboo with different cutting bandwidths (3m, 5m, 8m, 12m, and 15m) was used as experimental materials. In this instance, the results showed that the density, flexural strength, and shear strength of bamboo decreased first and then increased with the increase of harvesting bandwidth. The volume shrinkage first increased, then decreased, and then stabilized. The research also showed that the total dry density, basic density, flexural strength, and compressive strength of the strip-cut bamboo were lower than those of the control stand.

Despite interesting topics and analyses, I present some suggestions/questions that should be considered in my opinion for this article:

-          The abstract is very extensive (386 words). He proposes to shorten mainly the part with the results.

-          The authors stated more citations (34) but in discussing the results were used only 10 references what is not at all bad but for extensive critical discussion mistakes more references are lacking. Authors must have more discussion with other similar articles. Please, add the missing discussion.

-          The introduction should outline why the process being analyzed is important to know. Each reference should be referred to at length. The authors cite references in whole sets, e.g., line 72.

-          Conclusion and Discussion are joined, it is not arranged suitable.  I would like to suggest making a separate conclusion as a chapter and connecting the discussion with the results. It will be clearer.

Author Response

For research article

Response to Reviewer Comments

1. Summary

Thank you very much for taking the time to review this manuscript. Please find the detailed responses below and the corrections highlighted in the re-submitted files.

2. Point-by-point response to Comments and Suggestions for Authors

Comments 1: The abstract is very extensive (386 words). He proposes to shorten mainly the part with the results.

Response 1: The results section was shortened, the conclusion was added, and the number of words was changed to 301 words

Comments 2: The authors stated more citations (34) but in discussing the results were used only 10 references what is not at all bad but for extensive critical discussion mistakes more references are lacking. Authors must have more discussion with other similar articles. Please, add the missing discussion.

Response 2: The discussion section has been changed and references to articles on the effects of harvesting on timber properties have been added.

Comments 3: The introduction should outline why the process being analyzed is important to know. Each reference should be referred to at length. The authors cite references in whole sets, e.g., line 72.

Response 3: I made more changes to the introduction, and the L72 part was removed.

Comments 4: Conclusion and Discussion are joined, it is not arranged suitable.  I would like to suggest making a separate conclusion as a chapter and connecting the discussion with the results. It will be clearer.

Response 4: It separates the conclusion from the discussion.

Round 2

Reviewer 1 Report

Effects of different cutting widths on physical and mechanical properties of Bamboos pilosa under strip cutting

Habitat conditions and growth conditions significantly affect the properties of wood raw material obtained from different areas. The conditions for the growth of subsequent newly planted trees may depend on the method of felling the previous trees. In this context, the subject of the article can be considered interesting.

In relation to the previous version of the article, the authors made significant clarifications and took into account most of the changes indicated by the reviewer. The article has been supplemented and corrected. In my opinion, the article can be published in this version.

Author Response

We would like to thank the referee again for taking the time to review our manuscript.

Reviewer 2 Report

 L140: Please, specify the typical average diameter of the culm tested, to compare it with the international Standard for mechanical test (ISO 22157)

Author Response

 We would like to thank the referee again for taking the time to review our manuscript.

Q:L140: Please, specify the typical average diameter of the culm tested, to compare it with the international Standard for mechanical test (ISO 22157)

A: 30mm bamboo strips were used to make compressive and shear resistant specimens. The allowable error range of the specimen is ±1.0mm in length and ±0.5mm in width. We do not list the average size because of the strict specimen error. If you mean the DBH of the whole Moso bamboo plant, we mentioned above that the standard bamboo is selected for each forest, so we do not consider the influence of DBH, and only take the standard bamboo as the representative of the forest.

Thank you very much for your suggestion. After our discussion, we decided not to modify L140. Thank you for your understanding.